# Characterization of PEBP-like Genes and Function of *Capebp1* and *Capebp5* in Fruiting Body Regeneration in *Cyclocybe aegerita*

**DOI:** 10.3390/jof10080537

**Published:** 2024-07-31

**Authors:** Nan Tao, Bopu Cheng, Yuanhao Ma, Ping Liu, Hongmei Chai, Yongchang Zhao, Weimin Chen

**Affiliations:** 1Biotechnology and Germplasm Resources Institute, Yunnan Academy of Agricultural Sciences, Kunming 650223, China; tn1953@126.com (N.T.); cbp1997@sina.com (B.C.); jianxyjian@126.com (Y.M.); liuping0606@126.com (P.L.); chm621@aliyun.com (H.C.); yaasmushroom@aliyun.com (Y.Z.); 2Yunnan Provincial Key Lab of Agricultural Biotechnology, Kunming 650223, China; 3Key Lab of Southwestern Crop Gene Resources and Germplasm Innovation, Ministry of Agriculture, Kunming 650223, China

**Keywords:** *Cyclocybe aegerita*, fruiting body development, lamella redifferentiation, overexpression, PEBP, RNA interference

## Abstract

Phosphatidylethanolamine-binding proteins (PEBPs) play a crucial role in the growth and development of various organisms. Due to the low sequence similarity compared to plants, humans, and animals, the study of *pebp* genes in fungi has not received significant attention. The redifferentiation of fruiting bodies is exceedingly rare in fungal development. Hitherto, only a few studies have identified the *Capebp2* gene as being associated with this phenomenon in *Cyclocybe aegerita*. Thus, exploring the role of *pebp* genes in fruiting body development is imperative. In the present study, four *Capebp* genes (*Capebp1*, *Capebp3*, *Capebp4*, and *Capebp5*) were cloned from the AC0007 strain of *C. aegerita* based on genome sequencing and gene prediction. The findings indicate that the *pebp* family, in *C. aegerita*, comprises a total of five genes. Moreover, the sequence similarity was low across the five CAPEBP protein sequences in *C. aegerita*, and only a few conserved sequences, such as HRY and RHF, were identical. Expression analyses revealed that, similarly to *Capebp2*, the four *Capebp* genes exhibit significantly higher expression levels in the fruiting bodies than in the mycelium. Furthermore, overexpressed and RNA interference *Capebp1* or *Capebp5* transformants were analyzed. The results demonstrate that overexpression of *Capebp1* or *Capebp5* could induce the regeneration of the lamella or fruiting body, whereas the knockdown of *Capebp1* or *Capebp5* could lead to the accelerated aging of fruiting bodies. These findings highlight a significant role of *Capebp* genes in the generation of *C. aegerita* fruiting bodies and provide a foundation for further exploration into their involvement in basidiomycete growth and development.

## 1. Introduction

Phosphatidylethanolamine-binding protein (PEBP) was initially identified as a cytoplasmic soluble protein in the bovine brain and is known to have a strong affinity to phosphatidyl ethanolamine [1]. The structural domains conserved across the PEBP family in plants and animals consist of a large beta-sheet flanked by small beta-sheets on either side and two alpha-helices at the C-terminal. Also, the structure contains a highly conserved phosphate-binding pocket, which plays a crucial role in the functionality of the PEBP family [2,3]. The PEBP superfamily comprises over 400 members, markedly conserved across the evolutionary spectrum from bacteria to humans [4]. 

Mammals have four types of PEBP proteins: PEBP1, PEBP2, PEBP3, and PEBP4 [5]. The PEBP1 protein binds to RAF-1 and acts as an inhibitor of MEK phosphorylation and activation, thereby playing a critical role in cell signaling [6]. On the other hand, PEBP4 shows a significant involvement in various biological processes, such as cell membrane synthesis, myoblast differentiation, neural development, and tumorigenesis [7]. In normal tissues, PEBP4 is predominantly expressed in the cardiac, pulmonary, prostatic, and thyroidal compartments of mammals, with minor expression in the hepatic, colonic, cutaneous, adrenal cortical, and hematopoietic niches [8]. Notably, PEBP4 expression is elevated across all neoplastic tissues [9]. According to the genome sequence, in angiosperms, PEBPs can be classified into three major categories: FLOWERING LOCUS T (*FT*)-like genes, MOTHER OF FT AND TFL1 (*MFT*)-like genes, and TERMINAL FLOWER 1 (*TFL1*)/CENTRORADIALIS (*CEN*)-like genes [10]. In angiosperms, the *pebp* gene family has been identified as a major regulatory factor in a wide range of developmental processes, including flower formation, fruit set, vegetative growth, stomatal control, tuberization, and branching in perennials [11,12,13]. The *Arabidopsis thaliana* genome consists of six genes in the *FT/TFL1* family. The genes *FT*, TWIN SISTER OF FT (*TSF*), and *MFT* initiate the flowering process, while the BROTHER OF FT AND TFL1 (*BFT*), *TFL1*, and ARABIDOPSIS THALIANA CENTRORADIALIS (*ATC*) exert an opposite effect by suppressing the formation of flowers [14]. Typically, florigen-like proteins such as *FT*-like genes function as activators to facilitate the process of flowering and the formation of underground storage organs, whereas antiflorigen-like molecules such as TFL1/CEN-like proteins inhibit both flowering and underground storage organ development [15,16]. Conversely, gymnosperms possess only *FT*-like genes and *TFL1*-like genes. The *TFL1*-like gene plays a crucial role in the reproductive development of gymnosperms. *GymFT1* and *GymFT2*, the *FT*-like genes, regulate the growth rhythm and sexual development pathways in gymnosperms. However, when expressed in *Arabidopsis thaliana*, spruce’s *FT*-like and *TFL1*-like genes inhibit flowering, suggesting the origin of the functional evolution of *FT*-like genes from basal angiosperms [17,18]. Phylogenetic analyses have revealed the independent evolution of *FT*-like genes in gymnosperms and angiosperms [19].

Although the function of the PEBP protein has been investigated extensively in various species, it has been overlooked in fungi due to its low sequence similarity and limited conservation compared to animals and plants. Although numerous PEBP-like proteins have been annotated in the genome maps of multiple fungal species, their functions have not been thoroughly studied. In a previous study, we identified a *pebp*-like gene, *Capebp2*, through genome and transcriptome data analysis of the basidiomycetes model organism *Cyclocybe aegerita*. Interestingly, overexpression of *Capebp2* in *C. aegerita* AC0007 strain induced cap redifferentiation, including intact fruiting bodies or partial lamella during fruiting development stage [20]. In the present study, the other four PEBP-encoding genes, *Capebp1*, *Capebp3*, *Capebp4,* and *Capebp5*, of *C. aegerita* AC0007 strain were cloned based on genome annotation. Expression analyses revealed that the expression levels of four CaPEBP proteins were elevated in the fruiting body compared to the mycelia. In this study, we primarily investigate the genes *Capebp1* and *Capebp5*, and successfully generate their overexpression and RNA interference (RNAi) transformants. Our findings reveal that upregulation of *Capebp1* or *Capebp5* can induce regeneration of lamella or fruiting body, whereas downregulation can lead to premature aging of the fruiting bodies. These results provide novel insights into the functional role of PEBP in fungi.

## 2. Materials and Methods

### 2.1. Strains and Culture Conditions

*C. aegerita* dikaryotic strain AC0007 deposited in the National Germplasm Bank of Edible Mushroom (Yunnan) was used in this study. The mycelia were cultured on yeast extract peptone dextrose (YPD) medium (0.2% yeast extract, 0.2% peptone, 2% dextrose, and 1.5% agar) at 22 °C. The cultivation process for fruiting bodies involved nurturing the mycelia on a sawdust medium composed of 42% hardwood sawdust, 15% wheat bran, 1% gypsum, and 42% cottonseed shells. The protoplast regeneration medium contained 205 g of sucrose, 2 g of tryptone, 2 g of yeast extract, and 10 g of agar per liter of H_2_O. For the amplification of plasmids, the DH5a strain of *Escherichia coli* was cultured in Luria–Bertani medium with 100 mg/mL of ampicillin at 37 °C.

### 2.2. Capebp Genes Cloning, Sequence Alignment, and Phylogenetic Analysis

The AC0007 strain was cultured on YPD agar and the mycelium was harvested 9 days after cultivation at 25 °C. Genomic DNA was extracted from the mycelium using the fungal gDNA isolation kit (Biomiga, San Diego, CA, USA). Total RNA extractions from the mycelia and fruiting body were conducted using RNAiso Plus (TaKaRa, Dalian, China), followed by reverse transcription to obtain cDNA using iScript gDNA Clear cDNA Synthesis Kit (Bio-Rad, Hercules, CA, USA). The primers were designed based on the genomic information of the strain AC0007 (Appendix A), and the coding sequences of four *Capebp* genes (*Capebp1*, *Capebp3*, *Capebp4*, and *Capebp5*) were amplified by polymerase chain reaction (PCR). The resulting amplicons were ligated into the pMD19-T vector (TaKaRa, Dalian, China), and the plasmids were confirmed by sequencing (Tsingke Biotechnology Co., Beijing, China). The five *Capebp* genes (*Capebp1*, *Capebp2*, *Capebp3*, *Capebp4*, and *Capebp5*) are available in the GeneBank with the accession number of OR667028, OR667027, OR684566, OR684567, and OR667029, respectively. 

The five CaPEBP proteins were aligned using Clustal W (https://www.genome.jp/tools-bin/clustalw, 11 October 2023) [21], while the sequences were analyzed using ESPript3 (https://espript.ibcp.fr/ESPript/cgi-bin/ESPript.cgi, 11 October 2023) [22]. The PEBP protein sequences from plants, animals, fungi, and bacteria were retrieved from the NCBI GenBank database for comparative analyses. The phylogenetic tree was constructed by aligning the PEBP proteins across evolution using MAFFT version 7 [23], and the phylogenetic analysis was conducted based on maximum likelihood (ML) using RAxML-HPC2 v. 8.2.12 [24], as implemented in the CIPRES (https://www.phylo.org/portal2/login!input.action, 11 October 2023) portal [25]. The GTR+G+I model was utilized on 1000 rapid bootstrap (BS) replicates for all proteins. 

### 2.3. Vector Construction and Transformation

The construction and transformation of the overexpression/RNAi vector were conducted following the methodology outlined in a previous study [26,27]. The pATH vector encompasses the CaMV35S terminator, the *HygR* gene encoding hygromycin B phosphotransferase, and the Ca–actin promoter from *C. aegerita*. Together, these elements facilitate the overexpression of both *HygR* and the desired gene. After amplifying the *Capebp1* cDNA with the specified primer pair capebp1F/R, the products were inserted into the pATH vector between the Ca–actin promoter and T35s elements by ligating them within the *Eco*RV restriction endonuclease cleaved site. The RNAi vector of *Capebp1* was constructed using plasmid pAGH, according to a previously described method [27,28]. A 585-bp antisense fragment from the *Capebp1* gene, amplified with primers capebp1RiF and capebp1RiR (Appendix A), was inserted between the forward *C. aegerita* actin promoter and the reverse *C. aegerita* gpd promoter. The construction methods for the overexpression vector and RNAi vector of gene *Capebp5* are identical to those described above. To generate the overexpression vector, the cDNA sequence of *Capebp5* was amplified using the primers capebp5F and capebp5R. For constructing the RNAi vector, a 328 bp antisense fragment of the *Capebp5* gene was amplified with the primers capebp5RiF and capebp5RiR (Appendix A).

The plasmid with the pATH-pebp1, pAGH-pebp1, pATH-pebp5, and pAGH-pebp5 constructs was amplified in DH5α cells. As mentioned before, we utilized the PEG-mediated method for genetic transformation. The lysis buffer utilized for protoplast generation comprises 1% cellulase R-10 (Yakult Pharmaceutical Industry Co., Ltd., Tokyo, Japan), 1% lysozyme (Biktak BioTech, Beijing, China), and 1% lysozyme (Sigma–Aldrich, St. Louis, MI, USA). These enzymes were dissolved in a mannitol buffer at a concentration of 0.6 M. Subsequently, the protoplasts were washed using an STC buffer (consisting of 1.2 M sorbitol, 10 mM CaCl_2_, and 50 mM Tris-HCl, pH 7.0) to facilitate their combination with plasmid DNA for transformation. This reaction mixture was then added to a YPD regeneration medium composed of 20.5% sucrose, 0.2% tryptone, 0.2% yeast extract, and 1% agar, and a YPD medium containing 150 µg/mL hygromycin B was poured over the plates to select the transformants after 7 days.

### 2.4. Transformant Verification and Real-Time Quantitative PCR (qPCR)

The primer pair 19ha3 and 19ha4 was used to target the flanking area of the inserted fragment, and the colonies that exhibited resistance against hygromycin B were validated successfully. Next, the transformants were transferred into a YPD medium under hygromycin B selection at 150 µg/mL. All the strains and plasmids are maintained in our laboratory for future reference.

Total RNA extractions from the fruiting bodies of the transformants were conducted using RNAiso Plus (TaKaRa, Dalian, China), followed by reverse transcription to obtain cDNA using iScript gDNA Clear cDNA Synthesis Kit (Bio-Rad, Hercules, CA, USA). The glyceraldehyde-3-phosphate dehydrogenase (*gpd*) gene was used as the internal reference for qPCR. The primer pairs gpd qF/R, pebp1 qF/R, pebp3 qF/R, pebp4 qF/R, and pebp5 qF/R were used for the amplification of *gpd*, *Capebp1*, *Capebp3*, *Capebp4*, and *Capebp5*, respectively (Appendix A). PCR was performed in a volume of 20 mL using the iTaqTM Universal SYBR Green SuperMix in the CFX96 Real-Time PCR detection system according to the manufacturer’s instructions (Bio-Rad). The thermal cycling parameters for the PCR reactions were as follows: 95 °C for 20 s for initial polymerase activation, followed by 40 cycles at 95 °C for 5 s and 60 °C for 30 s. The primer pairs gpd qF/R, pebp1 qF/R, and pebp5 qF/R were used for the amplification of *gpd*, *Capebp1*, and *Capebp5*, respectively, in triplicates. The relative expression levels of the target genes were calculated using the 2^−∆∆CT^ method. 

### 2.5. Data Analysis

The experimental data analysis was performed using GraphPad Prism version 5 software, and the least significant difference (LSD) test was used to examine the significant differences among samples using a one-way analysis of variance (ANOVA). To ensure data precision, all experiments were conducted in triplicates.

## 3. Results

### 3.1. Sequence Analysis of Capebp Gene Family

Genome analyses revealed the presence of five *pebp* genes in the strain AC0007, arranged sequentially on one scaffold (Figure 1A). These were named *Capebp1*, *Capebp2*, *Capebp3*, *Capebp4*, and *Capebp5*, respectively, based on their sequence. gDNA and cDNA were used as templates to amplify the four *Capebp* genes based on genome annotation and sequences. The total length of *Capebp1* DNA was 787 bp and the gene consisted of three exons (E1–E3) and two introns (I1 and I2). Each of the DNA sequences of *Capebp3*, *Capebp4*, and *Capebp5* consisted of four exons (E1–E4) and three introns (I1–I3), with lengths of 849 bp, 800 bp, and 861 bp, respectively (Figure 1B).

The comparison revealed a low similarity among the five CaPEBP protein sequences in the strain AC0007; only conserved motifs, such as “DT”, “HRY”, “RHF”, etc., exhibited consistency. In terms of length, CaPEBP2 was longer, containing 299 amino acids, while the other four CaPEBP proteins had similar sequence lengths, with CaPEBP1 consisting of 194 amino acids, CaPEBP3 consisting of 210 amino acids, CaPEBP4 consisting of 209 amino acids, and CaPEBP5 consisting of 222 amino acids (Figure 1C).

The phylogenetic analysis of the PEBP protein sequences revealed distinct branches for species belonging to plants, animals, fungi, and bacteria. Notably, the PEBP proteins of plants and animals were clustered together, initially, before being grouped with the PEBP proteins of fungi, while the PEBP proteins of bacteria formed separate branches. The phylogenetic tree revealed that, among the five CaPEBP proteins of *C. aegerita*, CaPEBP5 had the highest dissimilarity compared to the other four, while CaPEBP1 and CaPEBP3 were highly similar (Figure 1D).

### 3.2. Expression Patterns of the Capebp Gene Family in Distinct Developmental Stages

In order to elucidate the expression patterns of four *Capebp* genes during different developmental stages of *C. aegerita*, we collected mycelia and the fruiting bodies for RNA extraction (Figure 2A). The expression levels of the four *Capebp* genes were quantified using qPCR, with the *gpd* gene as an internal reference. Our findings indicated that the mRNA expression levels of these four *Capebp* genes were significantly higher in the fruiting bodies compared to the mycelia (Figure 2B). The expression levels of *Capebp1* and *Capebp5* were approximately 30- and 66-fold higher in the fruiting body compared to the mycelia, respectively. Additionally, the expression levels of *Capebp3* and *Capebp4* were five and 16 times higher in the fruiting body than in the mycelia, respectively. 

### 3.3. Transformant Verification and Expression of Capebp1 and Capebp5

In order to study the effect of *Capebp1* on the developmental process, overexpressed constructs and RNAi transformants were obtained by constructing a pATH-pebp1 overexpression vector and a pAGH-pebp1 RNAi vector (Figure 3A, C). The positive transformants were identified by hygromycin B screening and PCR validation. Subsequently, these transformants were transferred to a medium containing hygromycin B for selection; five overexpression transformant strains (T7, T12, T16, T17, and T19) and three RNAi transformants (R4, R8, and R14) showed significant resistance to hygromycin B, whereas the growth of the wild-type strain was completely inhibited. Then, the transformants carrying the insertions were confirmed by PCR using primers 19ha3 and 19ha4 (Appendix A).

The overexpression vector pATH-pebp5 and RNAi vector pAGH-pebp5 for *Capebp5* were constructed and validated (Figure 3E, G). The screening of transformants was also conducted using two methods: hygromycin B resistance and PCR analysis (Appendix A). Ultimately, five overexpression transformants (T3, T4, T7, T9, and T12) and three RNAi transformants (R1, R13, and R16) were obtained for the next study.

The expression levels of *Capebp1* and *Capebp5* in their overexpressed transformants were higher compared to those of the wild-type strains (Figure 3B, F). The expression of *Capebp1* or *Capebp5* in the RNAi transformants was significantly downregulated (Figure 3D, H).

### 3.4. Effect of Capebp1 on the Development of Fruiting Bodies

In order to study the expression of *Capebp1* in the fruiting bodies of *C. aegerita*, we collected the fruiting bodies of the wild type and transformants for RNA extraction, followed by reverse transcription into cDNA. *Gpd* was used as an internal reference gene, and the expression of *Capebp1* was detected in the wild type and in the transformants by qPCR. The results showed a significant upregulation of the *Capebp1* gene in the overexpressed transformants of the fruiting bodies compared to the wild type (Figure 4A). Conversely, the expression of *Capebp1* was significantly downregulated in RNAi transformants compared to the wild type (Figure 4B). Compared to the wild type, the lamellae of *Capebp1*-overexpressed transformants exhibited varying degrees of redifferentiation (Figure 4C), such lamellae growing in the middle or edge of the cap. The morphology of the regenerated lamellae varied, including fully regenerated lamellae T7, T16, T17, and T19, as well as partially differentiated tissues in T12 (incomplete cap). During the same developmental cycle, the RNAi transformants exhibited signs of premature aging compared to the wild-type strain (Figure 4C). They can develop into complete fruiting bodies, but both mature and young mushrooms showed wilting traits.

### 3.5. Effect of Capebp5 on the Development of Fruiting Bodies

We further investigated the expression of *Capebp5* in the fruiting bodies of *C. aegerita* using the same method as for *Capebp1*. The results show that the *Capebp5* gene was significantly upregulated in the overexpressed transformants of the fruiting bodies compared to the wild type (Figure 5A, B). Compared to the wild type, the *Capebp5* overexpressed transformants exhibited redifferentiation in the lamellae or a complete fruiting body (Figure 5C). A complete fruiting body, consisting of a stalk and cap, developed on the cap of the transformant T3. However, T4, T7, T9, and T12 only exhibited an incomplete fruiting body or lamellae on the cap. RNAi transformants showed signs of premature aging when compared to the wild-type strain (Figure 5C).

## 4. Discussion

The formation and development of fungal fruiting bodies are regulated by a series of specific genes. In *Cordyceps militaris*, the overexpression of milR4 facilitates early fruiting body formation, whereas disruption impedes the development of fruiting bodies [29], a complex and enigmatic process. Mating-type pathway-related genes and transcription factors play pivotal regulatory roles in the development of fruiting bodies. The knockdown of the mating-type-related gene *Fvclp1* in *Flammulina velutipes* leads to fewer fruiting bodies than the wild-type strain. Conversely, the overexpression of *Fvclp1* can significantly enhance fruiting body formation [30]. The transcription factor *FvHmg1* plays a negative regulatory role in the development of fruiting bodies in the winter mushroom *F*. *velutipes* [31]. The production of the fruiting body in *Cordyceps militaris* has been shown to become impaired upon knockout of the flavohemoprotein-like gene *Cmflip* [32]. The transcriptional linkage among the three blue-light receptor genes, *CmWC-1*, *CmCRY-DASH*, and *Cmvvd*, in *Cordyceps militaris* regulates the development and secondary metabolism of the fruiting bodies [33]. The inhibition of the glutathione peroxidase (*GPX*) gene through RNAi can significantly impact mycelial growth and fruiting body development in *Hypsizygus marmoreus* [34]. There are numerous instances of specific genes regulating the development of fruiting bodies; however, apart from the case where overexpression of *Capebp2* induces cap redifferentiation in *C. aegerita* fruiting bodies [20], no studies have reported this phenomenon. The observation of fruiting bodies regeneration in this study resulting from the overexpression of *Capebp1* or *Capebp5* is also an uncommon occurrence. 

The *pebp* gene is a pivotal regulatory gene for growth and development and has been extensively and comprehensively investigated across humans, animals, and plants. PEBP4 impacts tumor survival rates by regulating AKT, MAPK, SHH, and other signaling pathways in human. Thus, investigating the expression of PEBP4 in tumors can offer novel insights and targets for preventing and treating malignancies [7]. Geophytes, such as potatoes and onions, exhibit a unique dual reproduction system encompassing vegetative and sexual propagation. Recent studies have shed light on the pivotal role of the *pebp* gene family in orchestrating the intricate molecular cascade underlying these two distinct modes of reproduction in the geophytes. In potatoes, the *pebp* gene family member *StSP3D* triggers flowering through long-distance migration to apical meristem, while *StSP6A* promotes tuber formation by migrating to the stolon [35,36]. *StSP5G* acts as an inhibitor of *StSP6A* [35], whereas another PEBP gene, *StCEN*, has been identified as a negative regulator of both flower development and tuber formation [37]. In the onion, the formation of bulbs is inhibited by a short photoperiod through the expression of *AcFT4*. However, once the critical long photoperiod is surpassed during the summer, bulb development is promoted due to the downregulation of *AcFT4* and upregulation of *AcFT1*. Overwintering and vernalization fulfill the requirement for a cold environment, resulting in the upregulation of *AcFT2*, which subsequently induces flowering in the following year [38]. The functional significance of the six *pebp* genes in *Arabidopsis thaliana* has been investigated extensively throughout its life cycle. For photoperiod flowering, the regulation of FLOWERING LOCUS T (FT) transcription is effectuated via two tightly controlled mechanisms. The first mechanism entails the activation of FT through the nuclear factor (NF)–CO module, while the second mechanism encompasses the epigenetic silencing of FT transcription mediated by polycomb groups during adverse flowering conditions. Subsequently, the FT is loaded into the neighboring sieve elements and transported via the phloem to the shoot apical meristem. The transportation of FT to the floral anlagen, followed by its interaction with the bZIP transcription factor FD and 14-3-3, initiates the expression of floral identity genes, such as *LEAFY* and *AP1*, thereby triggering flower primordium formation. Conversely, the interaction between FD and TFL1 inhibits FP formation while inducing the formation of leaf primordium. Cytokinins trigger the floral transition under short-day conditions via the upregulation of TSF transcription. Intriguingly, TSF exerts an inhibitory effect on kinase activity and can bind with fructose phosphorylase fructokinase 6 (FRK6), impeding its kinase activity [39]. Furthermore, FT and TSF are crucial in accelerating flowering induced by adjacent factors and have an impact on stomatal opening. Moreover, by regulating flowering time, FT and TSF influence collateral growth. The overlapping functions of FT and TSF aid the determination of inflorescence architecture [40]. The PIF1/SOM pathway has been shown to facilitate the expression of MFT and suppress seed germination under far-red-enriched light [41]. The 14-3-3 proteins have been described in eukaryotic organisms from fungi to humans, and their function is well understood in fungi. Although they share highly conserved protein sequences, 14-3-3 proteins seem to play different roles in different fungi, such as acting as the regulation factor for G2/M transition in *Ustilago maydis*, G1/S transition in *Saccharomyces cerevisiae* [42], and for the vegetative growth in *Candida albicans* [43]. Whether it interacts with PEBP, as in plants, it is still unknown. The regulation pathway of PEBP proteins involved in fungi should be uncovered in further works.

During mushroom production, some deformed mushrooms are often discovered, a phenomenon which is often considered to be related to environmental factors [44], with viruses also possibly contributing to deformed mushrooms production [45]. In *Lentinula edodes*, some fruiting bodies with abnormal margins can be found, but the deformity rate has been shown to decrease significantly over time in the same cultivation bag [44]. However, studies have assessed the fruiting body development and found that the phenomenon of fruiting body redifferentiation is exceedingly rare; hitherto, only the *Capebp2* gene in *C. aegerita* has been analyzed. The link between the *pebp* genes and different redifferentiation tissues, including intact fruiting bodies and lamellae, was further verified in this study, highlighting the importance of *pebp* genes in the fungal development. 

The cloning of four *Capebp* genes (*Capebp1*, *Capebp3*, *Capebp4*, and *Capebp5*) was successful in this study, resulting in the identification of a total of five *Capebp* genes within the *pebp* gene family of *C. aegerita*, including the previously known *Capebp2* gene. A comparative analysis among these five CaPEBP proteins and known PEBP proteins in plants and animals revealed significant differences, indicating a low level of similarity. The expression analysis revealed that the expression levels of all *Capebp* genes in the fruiting body of the wild-type AC0007 strain were significantly higher compared to those in the mycelium, suggesting a crucial involvement of *Capebp* genes in the development of fruiting bodies. The overexpression of *Capebp5*, similarly to *Capebp2*, has been shown to induce two distinct types of fruiting body redifferentiation: one type involves the development of a new and fully formed fruiting body on the cap, while the other type leads to the formation of an irregular partial lamella without a stalk [20]. In this study, the overexpression of *Capebp1* only led to the formation of lamellae. Therefore, we have concluded that the *Capebp* gene family may play a crucial role in fruiting body development in fungi, particularly in the redifferentiation process. In the meantime, further investigation is required to determine whether *Capebp2* and *Capebp5* play more prominent roles in the development of the fruiting body compared to *Capebp1*. Although the RNAi transformants of *Capebp1* or *Capebp5* can lead to premature aging, the underlying mechanism of this phenomenon remains unknown. Additionally, although the function of *Capebp3* and *Capebp4* remains to be discovered, their expression levels in the fruiting body are significantly higher compared to the low expression in the mycelium. It is hypothesized that these two genes may also play a pivotal role in fruiting body development. However, gene family members perceive environmental and biological factor signals, coordinating and balancing their expression, and, thus, affecting the development of fruiting bodies. Further research is required to gain a deeper understanding of the role of PEBP genes in fungal development.

## 5. Conclusions

The PEBP family of proteins is widely distributed across various species, playing diverse biological roles and exerting significant regulatory effects on the growth and development of organisms. However, research on PEBP in fungi is still at its nascent stage. In this study, genome mining revealed that five *Capebp* genes are present in the same scaffold in *C. aegerita*. The function of *Capebp1* and *Capebp5* was analyzed in this study. Overexpression of *Capebp1* or *Capebp5* can induce the formation of lamellae or of a fruiting body. Additionally, *Capebp1* or *Capebp5* silencing resulted in accelerated aging. These findings provide valuable insights for further investigations into the functional role of PEBP proteins in fruiting body development in fungi. However, an in-depth investigation is required to elucidate the generation and transport pathways of PEBP proteins, as well as to determine the targets and the underlying regulatory mechanisms characteristic of fruiting body development. Future studies should aim to explore the response to the environment, functional characteristics, and regulatory pathways of *pebp* gene in fungi.

## Figures and Tables

**Figure 1 jof-10-00537-f001:**
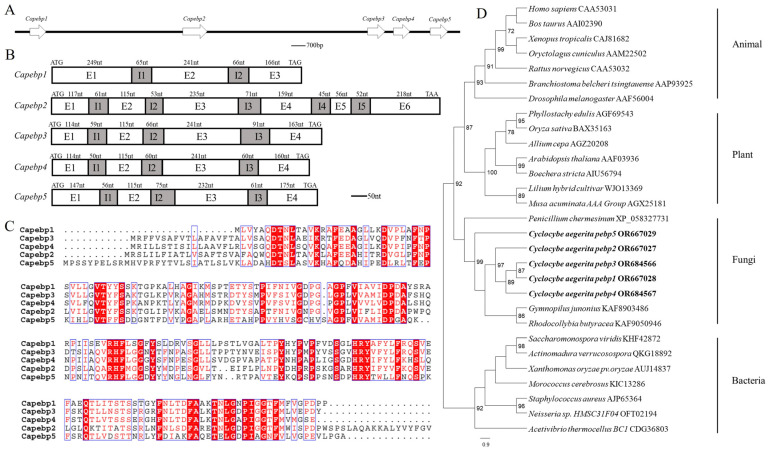
Sequence analysis of the *Capebp* gene family in *Cyclocybe aegerita*. (**A**) The *Capebp* genes are arranged in a sequential order on a scaffold. (**B**) Sequence structure analysis of five *Capebp* genes. *Capebp1* consists of three exons (E1–E3) and two introns (I1 and I2). *Capebp2* is composed of six exons (E1–E6) and five introns (I1–I5). *Capebp3*, *Capebp4*, and *Capebp5* each contain four exons (E1–E4) and three introns (I1-I3). (**C**) Partial amino acid alignment of CaPEBP protein sequences. (**D**) The phylogenetic tree illustrates the evolutionary relationships among five CaPEBP proteins and PEBP proteins from various species, encompassing plants, animals, fungi, and bacteria. PEBP sequences from species of plants, animals, fungi, and bacteria were obtained by blasting CaPEBP sequences against the NCBI GenBank database.

**Figure 2 jof-10-00537-f002:**
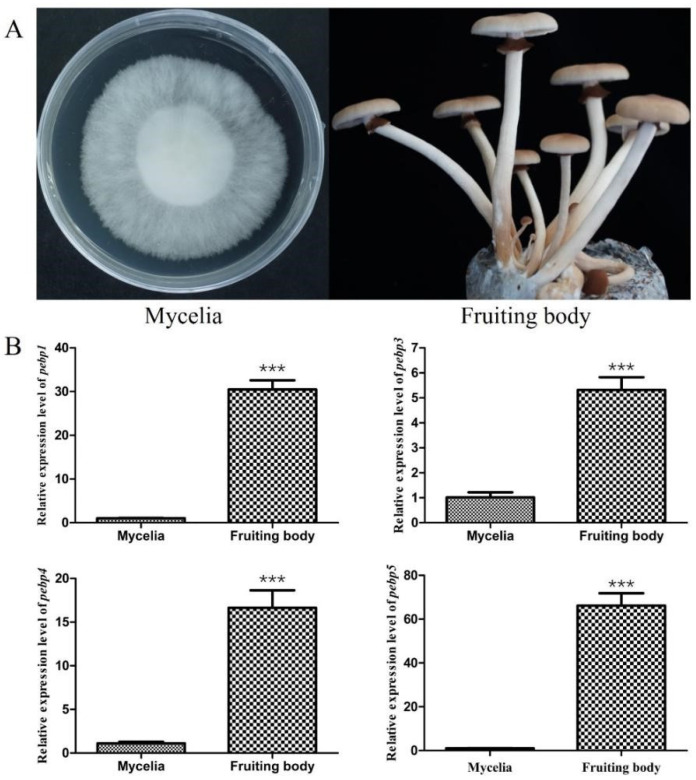
Expression analysis of four *Capebp* genes in the mycelia and fruiting bodies of *Cyclocybe aegerita*. (**A**) Mycelia and fruiting bodies of *C. aegerita* Ac0007 strain. (**B**) Real-time quantitative PCR analysis of four *Capebp* genes expression in the mycelia and fruiting bodies of *C. aegerita* Ac0007 strain. The asterisks (***) indicate significant differences compared with the mycelia (*p* < 0.0001, Student’s *t*-test).

**Figure 3 jof-10-00537-f003:**
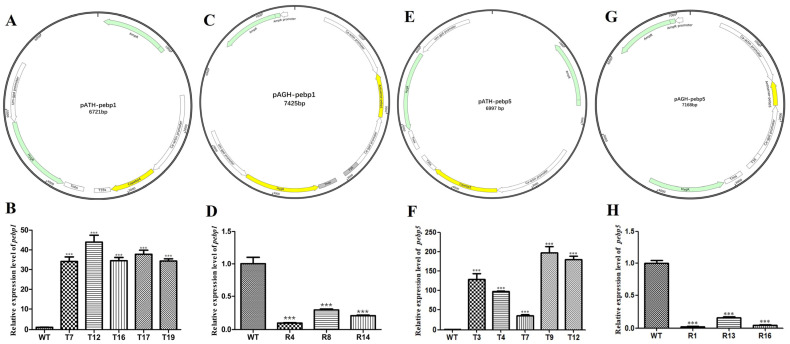
Construction of the *Capebp1* and *Capebp5* overexpression/RNAi vectors and expression analysis of the transformants. (**A**) Structure of the *Capebp1* overexpression plasmid pATH-pebp1. (**B**) The expression analysis of *Capebp1* overexpression transformants. (**C**) Structure of the *Capebp1* RNAi plasmid pAGH-pebp1. The antisense fragment of *Capebp1* was inserted in the site between the promoters Ca–actin and Ca–gpd. (**D**) Expression analysis of *Capebp1* RNAi transformants. (**E**) Structure of the *Capebp5* overexpression plasmid pATH-pebp5. (**F**) Expression analysis of *Capebp5* overexpression transformants. (**G**) Structure of the *Capebp5* RNAi plasmid pAGH-pebp5. (**H**) Expression analysis of *Capebp5* RNAi transformants. The asterisks (***) indicate significant differences compared with the wild type (WT) based on *p* < 0.0001 (*t*-test) as the significance threshold.

**Figure 4 jof-10-00537-f004:**
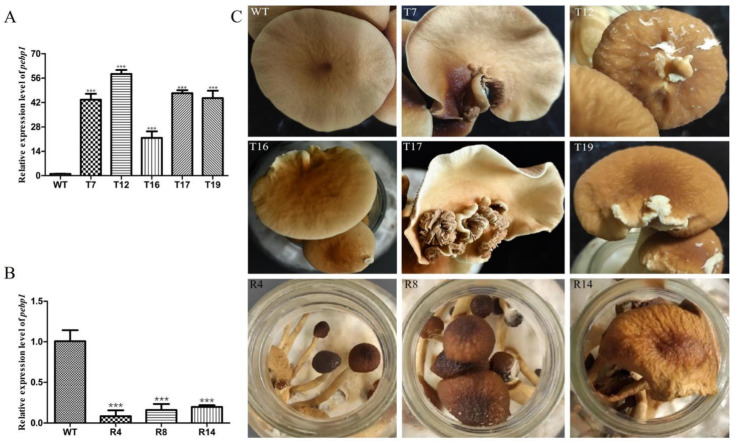
Overexpression/RNAi of *Capebp1* in *C. aegerita* has a significant influence on the pileus development. (**A**) Expression level of *Capebp1* in mature fruiting bodies of overexpression transformants. (**B**) Expression level of *Capebp1* in mature fruiting bodies of RNAi transformants. (**C**) Overexpression transformants (T7, T12, T16, T17, and T19) induce lamellar redifferentiation. The development of RNAi transformants (R4, R8, and R14) is not fully accomplished and senescence occurs rapidly. The asterisks (***) indicate significant differences compared with the wide type (WT) (*p* < 0.0001, Student’s *t*-test).

**Figure 5 jof-10-00537-f005:**
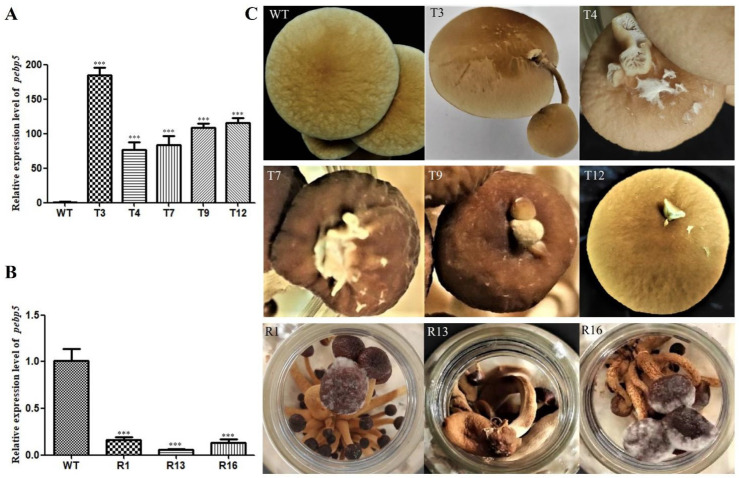
Overexpression/RNAi of *Capebp5* in *C. aegerita* has a significant influence on pileus development. (**A**) Expression level of *Capebp5* in mature fruiting bodies of overexpression transformants. (**B**) Expression level of *Capebp5* in mature fruiting bodies of RNAi transformants. (**C**) Overexpression transformants (T3, T4, T7, T9, and T12) induce lamellar or fruiting body redifferentiation. The development of RNAi transformants (R1, R13, and R16) is not fully accomplished and senescence occurs rapidly. The asterisks (***) indicate significant differences compared with the wide type (WT) (*p* < 0.0001, Student’s *t*-test).

## Data Availability

The original contributions presented in the study are included in the article/Appendix A, further inquiries can be directed to the corresponding author.

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
