# Peer review of "Characterization of PEBP-like Genes and Function of *Capebp1* and *Capebp5* in Fruiting Body Regeneration in *Cyclocybe aegerita"

_jof, 2024, doi:10.3390/jof10080537_

Round 1

Reviewer 1 Report (Previous Reviewer 2)

The resubmitted manuscript on the function of PEBP proteins in Cyclocybe has been improved by the additional functional characterization of another homologue, Capebp5. The up- or downregulation of these genes results in detectable phenotypes. Given however the fact that the conclusions are similar to the ones obtained from the study of the other Capebp’s characterized so far, a more precise discussion of these results is at least necessary in order to highlight novelty. In this context, please avoid repeating the same information in the discussion and the conclusion sections and consider rewriting-restructuring these parts, and including some more speculation on the actual function of five genes in Cyclocybe.

I do not have any more detailed comments.

Author Response

Reviewer 2 Report (New Reviewer)

The objective of the manuscript is to investigate the role of phosphatidylethanolamine-binding proteins genes in fruiting body development in fungus Cyclocybe aegerita. This study is a sequel to a previous study that identified the role of one of the 5 pebp genes. This paper reports new data about fungal development and the presented experiments seem to be well conducted. I find this research article suitable for publishing after minor correction.

I find this research article suitable for publishing after minor correction:

Line 213. The first line of the 3.2. section of the Results states that Capebp2 exhibits differential expression patterns between mycelia and fruiting bodies (Fig. 2A). However, Figure 2А depicts only photographs of mycelia and fruiting bodies of C. aegerita strain. It's confusing. In fact, Capebp2 expression is published in the previous author’s paper [20] that they refer to in the Introduction and Discussion sections. Rephrase this paragraph so that the reader is not looking for Capebp2 gene expression data in the Fig. 2A.

Author Response

This manuscript is a resubmission of an earlier submission. The following is a list of the peer review reports and author responses from that submission.

Round 1

Reviewer 1 Report

The article “Characterization of PEBP-like encoding genes and function of Capebp1 in lamella regeneration in Cyclocybe aegerita” deals with the role of pebp genes in the generation of C. aegerita fruiting bodies. The topic is very current and reveals the influence of the mRNA expression levels of Capebp genes over the extremely rare phenomenon of fruiting body redifferentiation. The research goal is clearly defined and also very well planned and executed experimentally. The results are convincing and clearly presented. Furthermore, the manuscript is well written and contains a sufficient number of literature references. Overall, I recommend the work for publication in present form.

N/A   

Author Response

Dear Reviewer:

Thanks for your comments. Next, we will improve the quality of the paper.

Reviewer 2 Report

Nan Tao and colleagues present an article on the identification and initial characterization of four genes encoding phosphatidylethanolamine-binding proteins (PEBP) in Cyclocybe aegerita, namely Capebp1, Capebp3 Capebp4, and Capebp5. Expression analysis revealed that the four Capebp genes exhibited higher expression levels in fruiting bodies and that overexpression of pebp1 induces regeneration of lamella, whereas knock- down of Capebp1 led to accelerated aging of fruiting bodies. Although these results are interesting, in the context of the data presented in a previous study by the authors in the same journal (https://doi.org/10.3390/jof9060657), another Capebp gene, Capebp2, is characterized, drawing similar conclusions related to its function. In my opinion, in its current form, the current manuscript does not provide a major advancement to the scientific literature.

Line 80: Capebp2 should be Capebp3

Table 1 could be sent to the supplement

Author Response

Dear Reviewer:

We would like to thank the reviewer for giving us constructive suggestions which would help us to improve the quality of the paper. Here we revised the manuscript with the title “Characterization of PEBP-like encoding genes and function of Capebp1 in lamella regeneration in Cyclocybe aegerita” (ID: jof-2923301) according to the reviewer’s comments. 

Sincerely yours,
All authors

Reviewer 3 Report

The manuscript “Characterization of PEBP-like encoding genes and function of Capebp1 in lamella regeneration in Cyclocybe aegerita” (jof-2923301) describes the analysis of four proteins belonging to the same family (previously the authors had already published the analysis of the fifth protein of that family) and their relationship with tissue differentiation in the model fungus Cyclocybe aegerita. The authors continue previous works and show a good handling of the methodology. Even so, there are two worrisome details regarding the independent reproducibility of their results:

1) The fungal strain is not accessible: "C. aegerita dikaryotic strain AC0007 deposited in our laboratory" (Line 90). Most journals request that the working species are also deposited in separate collections, so that the results could be reproducible by other authors. I do not know if this is the case for the Journal of Fungi.

2) The sequences are not deposited in any database: "The sequences of five Capebp genes were submitted to the NCBI database" (Line 110). OK, but they were not deposited in that database, because they do not exist in it. As discussed above for the fungal strain, the sequences should be deposited in accessible databases. The authors must deposit them and provide accession numbers. I consider this point indispensable.

The English language must be reviewed throughout the document. As examples of this, I would highlight:

- Lines 21-22. "The findings indicated that the inclusion of Capebp2 comprised a total of five pebp family genes in C. aegerita".

- Line 43. "Mammals consist of four types of PEBP proteins: PEBP1-4".

- Lines 136-137. "all solubilized at a concentration of 0.6 M in a mannitol buffer".

- Line 264: "fruition bodies".

Other aspects that may help improve the manuscript are listed below:

- Lines 2-3. The title lacks italics, booth in Capebp1 and Cyclocybe aegerita.

- Lines 31-32. Keywords are usually ordered alphabetically.

- Lines 102-103. The material and Methods section says, "Total RNA extraction from the mycelia was carried out using RNAiso Plus". The Results section (lines 201-203 ) mentions "In order to elucidate the expression patterns of four Capebp genes during different developmental stages of C. aegerita, we collected mycelia and fruiting bodies for RNA extraction. The expression levels of the four Capebp genes were quantified using qPCR, with the gpd…". All that information does not correspond to that section and should have been described with the RNA extraction in the Methods section. The same is true for the information contained in lines 236-238: "The mycelia of wild-type strains and transformants were collected for total RNA extraction, followed by cDNA synthesis through reverse transcription. The expression of capebp1 mRNA was determined using qPCR"

- Line 127. EcoRV is not fully typed in italics, only Eco.

- Lines 128-129. “A 585-bp antisense fragment”. Please change to “A 585-bp antisense fragment from Capebp1 gene”. It is only possible to understand what that is from the names of the primers...

- Lines 128-130. Plasmid pAGH includes a forward C. aegerita actin promoter and a reverse C. aegerita gpd promoter. This plasmid is intended to be used in C. aegerita, but already contains two genetic elements from the same fungus. They have not thought that this could be a reason for homologous recombination????

- Line 146: "we transferred ". Please, use the impersonal: the transformants were transferred.

Figure 3. Probably the whole figure is left out. A and D are interesting, they can go as supplementary material, B and E are unnecessary, and C and F are repeated in Figure 4.

Figure 4B. Why are there differences between the various over-expressed transformants? And the same question applies to the under-expressed transformants…

References.

Some references have words unnecessarily capitalized (e.g. lines 394-395, line 408, lines 415-416, lines 441-443, lines 447-448, and lines 449-450).

Some references do not have italics in words that require it (e.g. Arabidopsis in line 413, Gossypium hirsutum in line 416, and Capebp2 in line 427).

Author Response

(The authors gave the same response as above.)

Reviewer 4 Report

The authors have characterized PEBP-like genes in Cyclosybe aegerita and for one the 5 genes identified shown its potential function by over- and reduced expression. This showed that these genes are somehow involved in the (re)differentiation of fruiting bodies. There is indeed not much known on the function of these genes in fungi and the manuscript contributes thus to the knowledge of genes involved in redifferentiation and putative malformation in fruiting bodies sometimes seen. The authors do, however, not refer to such phenomenons.
The manuscript is an extension of the previous paper on CaPebp-2, and here the expression is describe of 4 other genes in this family.

The two papers reveal now the function of 2 of the 5 genes in this family. It is a pity that the authors did not examine the function of the other genes by transformation experiments. The authors could at least have used the previous CaPebp-2 transformants to examine the over- and reduced expression of CePebp-1 genes and see what the phenotype is when interfering with 2 of this gene family.

The authors have extensively described in the Introduction and the Discussion the function of this gene family in plant and animals. This repeated description shows that these genes have a wide ranges of roles in morphogenesis and other processes in these kingdoms that are different from hose in fungi. The authors do not discuss in what way the function in these kingdoms might be related to the function of these genes in fungi. Next to the description in plants and animals, the authors just repeat results in the Discussion paragraph and in the Conclusion paragraph (which is just a summary of the Discussion paragraph).  

Some minor items:
Line 43: “Mammals consist of four…”: Mammals have four..”
Line 44: RAF-1: mention the function of RAF-1
Line 86: “..will provide..”: “…provides..”
Line 176: “only conserved motives such as….”: Why only mentioning these three motives whereas the are at least 7?
Line 200 (Fig. 2A). The reference to tis figure at the end of the sentence suggest that this figure shows the differential expression of Capebp-2 in mycelia and fruiting bodies. The figure just shows the two stages in differentiation.
Lines 238-241: The words “Interesting” and “However” suggest that these results of the transformation experiments (over- and reduced expression) are a surprise. That is not the case . Leave out these words.
Paragraph 3.4 I would suggest to swap the phenotype description and gene expression. Show first that genes are up- or down regulated ad then the phenotypic effect.
Lines 243-244: The authors should describe here in more details what the phenotypic effects are.
Lines 261-286: Here the authors describe genes involved in the differentiation of mycelia into fruiting bodies. The expression interference of Cabebp genes induces redifferentiation, a process that is different than for genes involved in fruiting body induction. How relevant is the elaboration on these types of fruiting inducing genes/miRNAs on the function of the Capebp genes?
Lines 357-375: This paragraph of Conclusions is merely a repetition of what is mentioned in the discussion session and can be left out.

Author Response

(The authors gave the same response as above.)

Round 2

Reviewer 2 Report

I respect the authors decision to resubmit this article, however, I still think that in the context of the data presented in their previous study the manuscript does not meet the expectations with regards to originality.

not applicable

Reviewer 3 Report

The authors have resolved all the comments I made in my previous review.

 The authors have resolved all the comments I made in my previous review.

Reviewer 4 Report

The authors have processed my comments well.

No further comments